# OpenReview forum: "Federated In-Context Learning: Iterative Refinement for Improved Answer Quality"
_ICML.cc/2025/Conference — ICML 2025 poster_

### Official Review · Reviewer_dppu · 2025-03-13

**Overall Recommendation:** 3

**Summary:**

This paper proposes the Fed-ICL framework to harness the benefits of ICL while ensuring privacy preservation in sensitive settings, which is the first framework of iterative optimization of federated learning (FL) with a parameter-free communication scheme to enable iterative refinement of responses. The authors establish a theoretical foundation for Fed-ICL by analyzing its performance on a simplified single-layer Transformer model and conduct extensive experiments across a diverse set of QA tasks, which show their framework's effectiveness.

**Claims And Evidence:**

The evidence is convincing in general, but I think it might be better to show their framework's robustness to privacy attacks since they mention that they combine the efficiency property of ICL and the privacy robustness of FL.

**Essential References Not Discussed:**

NA

**Experimental Designs Or Analyses:**

My suggestions and concerns on the experimental part is in the Methods And Evaluation Criteria section. Thanks.

**Methods And Evaluation Criteria:**

The ablation study and the comparison with other methods on different datasets are comprehensive. One of my suggestion is the same as the previous section to prove the privacy ability somehow. Also, since these 2 datasets are in the area of QA, but right now more and more people care about the reasoning ability of LLMs, I wonder if this pipeline can still be effective in the reasoning tasks, which may make this work more impactful.

**Other Comments Or Suggestions:**

NA

**Other Strengths And Weaknesses:**

NA

**Questions For Authors:**

NA

**Relation To Broader Scientific Literature:**

This is an interesting work which combines 2 popular methods in the modern ML systems and I believe it can have a broader impact in the future.

**Theoretical Claims:**

The theoretical proofs are clear, smooth and rigorous.

---

> ### Author Rebuttal · Authors · 2025-04-01
>
> We sincerely thank the reviewer for the valuable time and effort in providing detailed feedback on our work.
>
> ---
> > **Q1:** The evidence is convincing in general, but I think it might be better to show their framework's robustness to privacy attacks since they mention that they combine the efficiency property of ICL and the privacy robustness of FL.
>
>
> **A1:**
> We thank the reviewer for the insightful question. We further conducted additional experiments to assess the privacy robustness of Fed-ICL. In particular, we evaluate the privacy robustness by Prompt Extraction Attacks [3], where each client generates responses to server queries using local knowledge, and we employ LLM to aim to reconstruct the original in-context examples using only the generated responses. Such a setup has been widely studied in previous works [1-4]. We list the prompt used for this reconstruction in Figure 4 ( https://anonymous.4open.science/r/Fed-ICL_ICML_rebuttal-2D96/Figure4.png ). We compared the reconstructed examples to the original ones, and the results of this comparison are presented in Figure 5 ( https://anonymous.4open.science/r/Fed-ICL_ICML_rebuttal-2D96/Figure5.png ).
> Our findings indicate that even a strong model like GPT-4o struggles to accurately recover the original in-context examples, demonstrating the robustness of the Fed-ICL framework against Memory Extraction Attacks.
>
>
> > **Q2:** Also, since these 2 datasets are in the area of QA, but right now more and more people care about the reasoning ability of LLMs, I wonder if this pipeline can still be effective in the reasoning tasks, which may make this work more impactful.
>
>
> **A2:**
> We thank the reviewer for the thoughtful feedback and for recognizing the contributions of our work. To the best of our knowledge, this study is the first to propose a framework that incorporates both in-context learning and federated learning with both theoretical and empirical support. We choose QA benchmarks such as MMLU and TruthfulQA following the previous works [6-11], which also study these two datasets in either federated learning or in-context learning.
> Motivated by the reviewer’s comment and inspired by prior reasoning studies [12-14],  we further evaluated the reasoning capabilities of Fed-ICL using a mathematical reasoning dataset.  We focused on the GSM-MC benchmark [5], a multiple-choice variant of GSM8K, enabling the straightforward use of majority-vote aggregation at the server side. In this extended experiment, we ran the experiments in a federated learning setting by partitioning the training dataset among three clients without overlap. Each client was equipped with the GPT-4o-mini model with in-context length 5. We randomly sampled 100 questions from the test dataset to serve as the server query data, following previous work [15]. We then evaluated Fed-ICL’s performance by measuring the accuracy of the generated answers on these questions.
>
> We show the results in Figure 1 ( https://anonymous.4open.science/r/Fed-ICL_ICML_rebuttal-2D96/Figure1.png ), which illustrates the performance progression of Fed-ICL and Fed-ICL-Free across communication rounds. The observed improvements over successive rounds indicate the robustness and effectiveness of Fed-ICL in addressing complex reasoning tasks.
>
> [1] On the privacy risk of in-context learning. arXiv:2411.10512, 2024.
>
> [2] Extracting training data from large language models. USENIX Security, 2021.
>
> [3] Effective prompt extraction from language models. arXiv:2307.06865, 2023.
>
> [4] Extracting prompts by inverting llm outputs. arXiv:2405.15012, 2024.
>
> [5] Multiple-choice questions are efficient and robust llm evaluators. arXiv:2405.11966, 2024.
>
> [6] Openfedllm: Training large language models on decentralized private data via federated learning. KDD, 2024.
>
> [7] Fedbiot: Llm local fine-tuning in federated learning without full model. Proc. KDD, 2024.
>
> [8] Understanding in-context learning from repetitions. arXiv:2310.00297, 2023.
>
> [9] Symbol tuning improves in-context learning in language models. arXiv:2305.08298, 2023.
>
> [10] Long-form factuality in large language models. arXiv:2403.18802, 2024.
>
> [11] Large language models are human-level prompt engineers. ICLR, 2022.
>
> [12] Gsm-symbolic: Understanding the limitations of mathematical reasoning in large language models. arXiv:2410.05229, 2024.
>
> [13] Large language models as analogical reasoners. arXiv:2310.01714, 2023.
>
> [14] A careful examination of large language model performance on grade school arithmetic. NeurIPS 37, 2024: 46819–46836.
>
> [15] Improving factuality and reasoning in language models through multiagent debate. ICML, 2023.
>
> [16] Few-shot In-context Learning on Knowledge Base Question Answering. ACL, 2023.
>
> [17] Fedmatch: Federated learning over heterogeneous question answering data. CIKM, 2021.

---

### Official Review · Reviewer_KbyF · 2025-03-13

**Overall Recommendation:** 3

**Summary:**

This paper introduces Fed-ICL, a framework that enhances in-context learning (ICL) for QA tasks. Specifically, Fed-ICL leverages iterative interactions between clients and a central server, progressively refining responses while maintaining low communication costs (by transmitting the context). The authors provide theoretical convergence guarantees and demonstrate strong performance on standard QA benchmarks.

**Claims And Evidence:**

Yes

**Essential References Not Discussed:**

NA

**Experimental Designs Or Analyses:**

Yes

**Methods And Evaluation Criteria:**

Yes

**Other Comments Or Suggestions:**

NA

**Other Strengths And Weaknesses:**

**Strengths**

1. To the best of my knowledge, this work presents a novel idea by combining federated learning and in-context learning.

2. The paper is well-written.

3. The method is effective and supported by comprehensive experiments.

**Weaknesses**

1. The paper focuses only on the QA dataset, and it is unclear whether it can generalize to more challenging tasks.

2. Do the methods work for advanced models or even reasoning models? What if reasoning models do not require in-context exemplars?

**Questions For Authors:**

What do the refined in-context exemplars look like, and how are they different from the original in-context exemplars?

**Relation To Broader Scientific Literature:**

The method is novel compared to related works.

**Theoretical Claims:**

Yes

---

> ### Author Rebuttal · Authors · 2025-04-01
>
> We sincerely thank the reviewer for the valuable time and effort in providing detailed feedback on our work.
>
> ---
> > **Q1:**  The paper focuses only on the QA dataset, and it is unclear whether it can generalize to more challenging tasks.
>
> **A1:**
> First, we would like to highlight that this work is the first to integrate in-context learning and federated learning, supported by both theoretical analysis and empirical validation. Our choice of QA benchmarks, including MMLU and TruthfulQA, aligns with prior work [2–7] in either federated or in-context learning. To explore more challenging tasks, and inspired by prior reasoning studies [8–10], we further evaluated Fed-ICL on GSM-MC [1], a multiple-choice variant of GSM8K for mathematical reasoning. We partitioned the training data across three non-overlapping clients, each using GPT-4o-mini with an in-context length of 5. Following prior work [11], we randomly sampled 100 test questions.
> Results are shown in Figure 1 (https://anonymous.4open.science/r/Fed-ICL_ICML_rebuttal-2D96/Figure1.png), which illustrates the performance progression of Fed-ICL and Fed-ICL-Free over communication rounds. The consistent improvements highlight the robustness and effectiveness of Fed-ICL on more complex reasoning tasks.
>
> > **Q2:** Do the methods work for advanced models or even reasoning models?
>
> **A2:**
> We believe our framework is broadly applicable to advanced and reasoning-oriented models, as it makes no assumptions about model architecture or output format. While we did not initially include experiments with such models, we conducted additional experiments inspired by the reviewer’s suggestion to assess Fed-ICL’s reasoning ability. As detailed in **A1**, our results on the GSM-MC benchmark demonstrate that Fed-ICL performs effectively on mathematical reasoning tasks, supporting the generality and robustness of our approach.
>
> > **Q3:** What if reasoning models do not require in-context exemplars?
>
> **A3:**
> We argue that in-context exemplars often remain beneficial, even for advanced models [14–15]. To support this, we conducted additional experiments comparing four variants: Fed-ICL, Fed-ICL-Free, Fed-ICL-GT, and a baseline LLM without in-context exemplars, as suggested by the reviewer. We evaluated both a standard backbone (LLaMA-3.1-8B) and a more advanced one (GPT-4o-mini).
> Consistent with our main paper, we used accuracy on MMLU and BERTScore on TruthfulQA. Results are shown in Table 1 (https://anonymous.4open.science/r/Fed-ICL_ICML_rebuttal-2D96/Table1.png) and Table 2 (https://anonymous.4open.science/r/Fed-ICL_ICML_rebuttal-2D96/Table2.png). Across both backbones and benchmarks, the baseline LLM without exemplars consistently underperforms the exemplar-based variants. This trend underscores the effectiveness of Fed-ICL and reaffirms the utility of in-context exemplars—even for more capable models.
>
>
> > **Q4:** What do the refined in-context exemplars look like, and how are they different from the original in-context exemplars?
>
> **A4:**
> We show additional examples from TruthfulQA in Figures 2 ( https://anonymous.4open.science/r/Fed-ICL_ICML_rebuttal-2D96/Figure2.png ) and Figures 3 ( https://anonymous.4open.science/r/Fed-ICL_ICML_rebuttal-2D96/Figure3.png )  to illustrate how the data evolves throughout federated learning. We observe that the server outputs become more and more closer to the ground truth. The client's outputs become increasingly professional and detailed. These demonstrate the benefit of our interactive process.
>
> ## References
>
> [1] Multiple-choice questions are efficient and robust LLM evaluators. arXiv:2405.11966, 2024.
>
> [2] OpenFedLLM: Training large language models on decentralized private data via federated learning. KDD, 2024.
>
> [3] FedBIoT: LLM local fine-tuning in federated learning without full model. KDD, 2024.
>
> [4] Understanding in-context learning from repetitions. arXiv:2310.00297, 2023.
>
> [5] Symbol tuning improves in-context learning in language models. arXiv:2305.08298, 2023.
>
> [6] Long-form factuality in large language models. arXiv:2403.18802, 2024.
>
> [7] Large language models are human-level prompt engineers. ICLR, 2022.
>
> [8] GSM-Symbolic: Understanding the limitations of mathematical reasoning in large language models. arXiv:2410.05229, 2024.
>
> [9] Large language models as analogical reasoners. arXiv:2310.01714, 2023.
>
> [10] A careful examination of large language model performance on grade school arithmetic. NeurIPS, 2024.
>
> [11] Improving factuality and reasoning in language models through multiagent debate. ICML, 2023.
>
> [12] Few-shot in-context learning on knowledge base question answering. ACL, 2023.
>
> [13] FedMatch: Federated learning over heterogeneous question answering data. CIKM, 2021.
>
> [14] Meta-in-context learning in large language models. NeurIPS, 2023.
>
> [15] Are emergent abilities in large language models just in-context learning? arXiv:2309.01809, 2023.

---

### Official Review · Reviewer_Y11y · 2025-03-17

**Overall Recommendation:** 3

**Summary:**

The paper proposes **Federated In-Context Learning (Fed-ICL)**, a framework for QA tasks that combines in-context learning and federated learning without transmitting model parameters. Fed-ICL enables clients to iteratively refine responses by sharing answers—not models—preserving privacy and reducing communication overhead. The authors provide theoretical guarantees of convergence and introduce **Fed-ICL-Free**, a variant for scenarios without labeled answers. Experiments on QA benchmarks show that Fed-ICL outperforms traditional FL and parameter-free methods, with ablation studies confirming the effectiveness of its components.

**Claims And Evidence:**

Figure 2 and Figure 3 present the main experiments of the paper. They compare different methods, including FL-based approaches and various parameter-free methods. Through these experiments, the paper demonstrates the effectiveness of the proposed method. Additionally, the paper conducts ablation studies to showcase the robustness of the method. Section 4 provides the theoretical proofs.

**Essential References Not Discussed:**

None

**Experimental Designs Or Analyses:**

Yes.

**Methods And Evaluation Criteria:**

To be honest, I’m not familiar with federated learning and haven’t done any research related to this area before. Therefore, I find it difficult to judge the novelty of the method proposed in the paper or evaluate its experimental setup. I’m unable to provide effective feedback on these aspects.

**Other Comments Or Suggestions:**

None

**Other Strengths And Weaknesses:**

None

**Questions For Authors:**

None

**Relation To Broader Scientific Literature:**

Federated Learning, In context learning.

**Theoretical Claims:**

Section 4.

---

> ### Author Rebuttal · Authors · 2025-03-31
>
> We sincerely thank the reviewer for their valuable time and thoughtful feedback. We also appreciate your kind acknowledgment of our contributions to both the experimental and theoretical aspects of the work in Claims and Evidence.

---

### Official Review · Reviewer_eC3B · 2025-03-23

**Overall Recommendation:** 3

**Summary:**

The paper introduces Fed-ICL, a novel framework that blends federated learning with in-context learning to tackle question-answering tasks in a privacy-preserving manner. Fed-ICL operates in a round-based manner, iteratively refining answer quality through client-server communication. The authors support their framework with a theoretical convergence guarantee based on a simplified single-layer linear self-attention model and provide extensive experimental evaluations on benchmarks such as MMLU and TruthfulQA.

**Claims And Evidence:**

Yes

**Essential References Not Discussed:**

No

**Experimental Designs Or Analyses:**

Yes

**Methods And Evaluation Criteria:**

Yes

**Other Comments Or Suggestions:**

None

**Other Strengths And Weaknesses:**

Strength: The paper makes a contribution by combining the ideas of federated learning and in-context learning. This integration is particularly valuable for applications with data privacy constraints and limited access to high-quality annotated examples. The inclusion of theoretical guarantees for convergence under a simplified linear model setting lends credibility to the proposed iterative refinement process and helps ground the empirical findings.

Weakness: The convergence guarantee is derived under a simplified linear self-attention model. While this is common for theoretical analysis, it remains an open question how these guarantees extend to more complex, fully nonlinear transformer architectures used in practice.

**Questions For Authors:**

None

**Relation To Broader Scientific Literature:**

Finding ans results

**Theoretical Claims:**

No

---

> ### Author Rebuttal · Authors · 2025-04-01
>
> We sincerely thank the reviewer for the valuable time and effort in providing detailed feedback on our work.
>
> ---
> > **Q1:** The convergence guarantee is derived under a simplified linear self-attention model. While this is common for theoretical analysis, it remains an open question how these guarantees extend to more complex, fully nonlinear transformer architectures used in practice.
>
> **A1:**
> We appreciate the reviewer’s suggestion and agree that extending our theoretical analysis to more general transformer architectures is an important direction. In our current work, we establish convergence guarantees for a simplified linear self-attention model to provide theoretical insights that support our empirical findings. This modeling choice is in line with several recent works aiming to understand transformers under linear assumptions (e.g., [1], [2]). Extending the analysis to fully nonlinear transformer architectures is a challenging and independent research direction, which goes beyond the scope of our current work.
>
> While analyzing fully nonlinear transformer architectures remains a significant challenge, we believe it is a promising and independent line of research. In particular, we conjecture that combining our framework with the recent results of [3], which show that an (L+1)-layer transformer can approximate L steps of in-context gradient descent (see their Section 3.5) could provide a pathway to deriving theoretical guarantees for federated in-context learning in more expressive models. However, we emphasize that such an extension is highly nontrivial and currently remains an open theoretical problem.
> We will incorporate this expanded discussion into the final version of the paper, as suggested.
>
>
> ## References
>
> [1] Transformers learn in-context by gradient descent. ICML, 2023.
>
> [2] Trained transformers learn linear models in-context. JMLR, 2024.
>
> [3] Transformers as statisticians: Provable in-context learning with in-context algorithm selection. NeurIPS, 2023.

---

### Decision · Program_Chairs · 2025-05-01

**Decision:**

Accept (poster)

**Comment:**

This paper proposed a federated learning algorithm (Fed-ICL) to improve the context data for in-context learning to improve QA task quality. Experiments show Fed-ICL outperforms baselines of federated learning (FedAvg) and LLM-debate, and theory based on existing ICL convergence extends to Fed-ICL for federated learning. The reviewers have a general consensus on acceptance with borderline scores of 3,3,3,3. Three out of four reviewers acknowledge reading the authors' response. However, reviewers also mention their low confidence on assessing this paper because they do not necessarily have expertise on both FL and ICL.

I also give the paper a quick read. The paper is well written and easy to follow. I agree with the reviewers on the contributions of combining FL and ICL, and appreciate the extensive experiment study with 2 datasets, four models, and many ablation baselines. However. I also have some concerns. A primary concern is the motivation of the paper, what is the practical usecase of the setting where there exist many local context datasets, and a query dataset without labels. As reviewer dppu mentioned, the privacy consideration needs to be clarified and justified. Highlighting discussions on how to map the heterogeneity (alpha for artificial heterogeneity in experiments) to practical considerations will also help.  A secondary concern is the comparison to previous work where baseline AFAIK either only use local context datasets, or only use server query dataset. This concern is secondary as the extensive ablation study helps justify the usefulness of the method. A minor request is to discuss related work such as CAN TEXTUAL GRADIENT WORK IN FEDERATED LEARNING? ICLR'25, which can be accessed since ICLR submission in Oct 2024.

In conclusion, this is a borderline paper. I appreciate the authors' efforts on drawing connection of FL to the LLM-debate settings, and the nice experiments in this setting. On the other hand, the draft can clearly be improved to address the concerns on motivation and experiments.